# Severe Radiation-Induced Lymphopenia Affects the Outcomes of Esophageal Cancer: A Comprehensive Systematic Review and Meta-Analysis

**DOI:** 10.3390/cancers14123024

**Published:** 2022-06-20

**Authors:** Dongjun Dai, Qiaoying Tian, Genhua Yu, Yongjie Shui, Hao Jiang, Qichun Wei

**Affiliations:** 1Department of Radiation Oncology, The Second Affiliated Hospital, Zhejiang University School of Medicine, Hangzhou 310009, China; daidongjunmed@zju.edu.cn (D.D.); 3150102475@zju.edu.cn (Q.T.); y18131229@zju.edu.cn (G.Y.); shui-yongjie@zju.edu.cn (Y.S.); 2Anhui Campus of the Second Affiliated Hospital, Zhejiang University School of Medicine, Bengbu 233000, China; 3Department of Radiation Oncology, The First Affiliated Hospital of Bengbu Medical College, Bengbu 233000, China

**Keywords:** esophageal cancer, radiation therapy, lymphopenia, pathologic complete response, survival outcomes, effective dose to the immune cells

## Abstract

**Simple Summary:**

Radiotherapy is as an important part of esophageal cancer (EC) treatment. However, it often causes severe radiation-induced lymphopenia (RIL). The aim of the current study was to evaluate the influence of severe RIL on the outcomes of EC. A systematic review and meta-analysis including 17 studies was performed. Our meta-analysis found that severe RIL was associated with a lower pathologic complete response rate and inferior overall survival and progression-free survival of EC patients. The lymphocyte nadir was found during 4–6 weeks after the start of radiotherapy. A series of dosimetric factors and clinical factors associated with RIL were summarized. Our results provide important evidence for the clinical application of radiotherapy. Minimizing the dosimetric risk factors, especially in patients with clinical risk factors, might benefit their outcomes. Our results might also offer clues for the strategy of combining radiotherapy and immunotherapy in EC patients.

**Abstract:**

The aim of the current study was to evaluate the influence of severe radiation-induced lymphopenia (RIL) on the outcomes of esophageal cancer (EC). A systematic review and meta-analysis was performed through the PRISMA guideline. Seventeen studies were included in the current systematic review, with eight included in the meta-analyses. Meta-analyses found that severe RIL was associated with lower pathologic complete response (pCR) rate (odds ratio (OR) = 0.44, 95% confidence interval (CI) = 0.30–0.66, I^2^ = 0%), inferior overall survival (OS) (hazard ratio (HR) = 1.50, 95% CI = 1.29–1.75, I^2^ = 6%), and worse progression-free survival (PFS) (HR = 1.70, 95% CI = 1.39–2.07, I^2^ = 0%) of EC patients. The lymphocyte nadir was found during 4–6 weeks after the start of radiotherapy. The leading dosimetric factors associated with severe RIL included larger PTV, higher dose to heart and body, and higher effective dose to the immune cells (EDIC). Clinical risk factors for RIL mainly comprised lower baseline ALC, higher tumor length and clinical stage, and distal EC. In conclusion, severe RIL might be associated with a lower pCR rate and worse OS and PFS of EC patients. Minimizing the dosimetric risk factors, especially in patients with clinical risk factors, might benefit their outcomes.

## 1. Introduction

Esophageal cancer (EC) is one of the most aggressive malignancies, with morbidity and mortality ranked seventh and sixth worldwide, respectively [1]. Approximately half of EC patients have cancer that extends beyond the locoregional confines of the primary site [2]. The 5-year overall survival (OS) of all EC patients is only about 20% in the United States [3]. Radiotherapy plays an important role in the management of EC, which serves as neoadjuvant chemoradiotherapy (CRT), followed by surgery or definitive CRT [4]. However, it often causes treatment-associated toxicities, including effects on host immunity such as lymphopenia [5].

Radiotherapy may play a primary role in the incidence of treatment-related lymphopenia in cancer patients [5], since lymphocytes are so radiosensitive that even a fraction of 2 Gy can kill half of the irradiated lymphocytes [6]. Radiation-induced lymphopenia (RIL) affects all lymphocyte subsets (CD^3+^, CD^4+^, CD^8+^ T cells, B cells, and natural killer cells) [7]. A study revealed that total lymphocyte counts did not change after neoadjuvant chemotherapy in non-small-cell lung cancer, while steep declines were found after the initiation of thoracic radiotherapy [8]. There are also studies wherein both radiotherapy and concurrent CRT significantly decreased the mean absolute number of lymphocytes compared to pretreatment levels, but no differences were detected in the characteristics of the lymphopenia induced by the two treatments in cervical cancer [9,10]. A study evaluated the radiation-induced hematological toxicity in patients with solid malignant tumors, which showed that at the end of radiotherapy, the patients with EC had the second lowest lymphocyte counts among common solid tumors, only better than head and neck cancer [11].

Previous studies found that RIL was associated with poor prognosis in several types of cancers [12]. Through multivariate analysis, meta-analyses showed that RIL was linked to worse overall survival (OS) in cancers such as lung cancer, brain cancer, pancreatic cancer, and head and neck cancer [13,14]. There are plenty of studies that showed that RIL was associated with worse OS [15,16,17,18], worse progression-free survival (PFS) [16,17,19], worse DMFS [20,21], worse local relapse-free survival (LRFS) [17], and worse disease-specific survival (DSS) [20] of EC patients. Previous studies also observed that severe RIL was related to a lower pathologic complete response (pCR) rate [19,22].

There was a previous meta-analysis that found that RIL was associated with inferior OS of severe solid tumors [13]. This meta-analysis included three EC studies [16,20,23] and found that the individual adjusted hazard ratio (HR) estimated in these studies was consistently larger than 1, and pooled HR demonstrated a trend toward statistical significance (HR = 1.46, 95% confidence interval (CI) = 0.95–2.23). It should be noted that this study only analyzed the association between RIL and OS of EC patients, and we found that there were some related studies that were not included in this meta-analysis [15,17]. Therefore, we performed this more comprehensive systematic review and meta-analysis. Our systematic review seeks to determine whether RIL is associated with poorer outcomes in EC patients and to summarize the risk factors for severe RIL in EC patients. The results of our study might provide valuable information for radiotherapy strategy in EC patients.

## 2. Materials and Methods

### 2.1. Literature Search

The current systematic review was performed according to the Preferred Reporting Items for Systematic reviews and Meta-Analyses (PRISMA) 2020 guidelines [24]. The literature was searched on the following databases: PubMed, Embase, and Cochrane Library. The search was updated to 4 May 2022. Appendix A lists the searching strategy of the current study. Our systematic review was registered at Open science (https://osf.io/prereg/ (accessed on 10 May 2022)) with a registration number of 10.17605/OSF.IO/HK7QU.

### 2.2. Inclusion and Exclusion Criteria

The inclusion and exclusion criteria were designed based on the population, intervention, comparison, outcome, and study design (PICOS) model (Appendix A). In detail, the target population was the EC patients who had radiotherapy as a part of their therapy strategy; the intervention was the EC patients with severe RIL; the comparison was the EC patients without severe RIL; the outcome was defined as the outcome data on the association between severe RIL and non-severe RIL, such as pCR and survival-related outcome data; and the study design type should be an observational study (either retrospective or prospective).

Included studies met the following criteria: (1) a cohort study which includes EC patients who had radiotherapy during their treatment; (2) recorded RIL information during or after radiotherapy; (3) contains RIL-related outcome data, such as pCR rate or survival outcome data.

The exclusion criteria of the current study were as follows: (1) a study with a cohort that has less than 10 patients, (2) written in a non-English language, (3) published before the year 2007.

### 2.3. Review Process and Data Extraction

The review process was performed according to the PRISMA 2020 flow diagram for new systematic reviews, only including searches of databases and registers [24] (Figure 1). The identified records from different databases were first screened by removing the duplicates. Potential related studies were collected after reading the titles and abstracts, and further assessed by the inclusion and exclusion criteria described above.

The following information from each included study was extracted: first author/publication year, region where the cohort was from, number of patients, tumor stage, age, histology, tumor location, median follow-up time, radiotherapy technique, dose plan, chemotherapy and surgery information, definition of severe RIL and data collection time, and severe RIL rate. The quality of these final selected studies was evaluated using the Newcastle-Ottawa Quality Assessment Form for Cohort Studies (NOS) [25]. In addition, we extracted the risk factors for potentially developing severe RIL in EC patients from the studies that used multivariate analysis as the calculating method. Furthermore, we extracted the data of absolute lymphocyte count (ALC) from studies which recorded them before and after the start of radiotherapy. The review process, data extraction, and quality assessment were performed by two independent investigators. Disagreements were further discussed and resolved by three reviewers.

### 2.4. Statistical Analyses

The R 4.0.5 software (http://www.R-project.org (accessed on 4 May 2022); ‘meta’ package) was used for meta-analyses. For the pCR rate, the pooled odds ratio (OR) and 95% CI were calculated. For survival outcomes, the pooled HR and 95% CI were calculated. The pooled analyses were performed through the generic inverse variance method by using the “metagen” function. Only studies with OR or HR calculated between two specified groups (a severe RIL group and a comparably non-severe RIL group) could be included in the meta-analysis. Otherwise, we only listed results as a part of the systematic review. For the studies including duplicate cohorts, the most recent, largest, or best-quality studies were selected for our meta-analysis. The meta-analysis was conducted only if there were equal or over three individual results from multivariate analysis. The statistical heterogeneity was evaluated by the I^2^ test. The random effect model was used in our meta-analysis since random- and fixed-effects models present similar results when heterogeneity is low [26]. If high heterogeneity occurred, a sensitivity analysis was performed to explore the potential source of heterogeneity by using the “metainf” function. The publication bias was assessed through Egger’s test by using the “metabias” function. Forest plots were used to visually display the results of individual studies and the synthetic results of the current meta-analysis. The curve plots representing the ALC tendency of EC patients who received CRT were plotted by using the “ggplot2” package. A two-sided *p* value of less than 0.05 was considered statistically significant.

## 3. Results

### 3.1. Results of the Review Process

As shown in Figure 1, there were 291 records identified after searching the databases. Among them, we removed 44 duplicated studies, and a further 209 studies by judging the titles and abstracts. The selected 38 studies were further assessed, and we excluded 4 studies with no RIL information, 3 studies which were not about EC, 1 study with participants who had not received radiotherapy, 1 study with duplicate data, and 12 studies with no information on outcome data. Finally, we included 17 retrospective studies [15,16,17,18,19,20,21,22,23,27,28,29,30,31,32,33,34] in our systematic review. The quality assessment results are listed in Appendix A. Relatively high quality was identified for all studies. There were 10 studies [15,16,17,18,19,20,21,22,23,34] which calculated the OR or HR on outcomes of EC patients between severe RIL and non-severe RIL groups, of which 3 studies [18,20,21] had the potential to have partial duplicate EC patients in their cohorts; therefore, we included the largest cohort [20] among them in our meta-analysis and kept the other 2 studies [18,21] in our systematic review.

As shown in Table 1, 17 studies were finally included in our systematic review, of which 8 studies [15,16,17,19,20,22,23,34] were included in the meta-analysis and 9 studies [18,21,27,28,29,30,31,32,33] were only included in the systematic review. The studies were published between the years 2017 and 2021. Most studies were from the United States (*n* = 7) and Eastern Asia (*n* = 9). The United States had more esophageal adenocarcinoma (EAC) and lower EC, while Eastern Asia had more esophageal squamous cell carcinoma (ESCC) and upper/middle EC. The majority of studies had concurrent chemotherapy with radiotherapy dose around 50.4 Gy and 1.8–2 Gy per fraction. Most studies collected the lowest ALC during the CRT and defined the Criteria for Adverse Events (CTCAE) grade 4 lymphopenia as the severe RIL. 

### 3.2. The Association between RIL and pCR of EC Patients

Four studies [19,22,33,34] reported the association between RIL and pCR of EC patients. Zhou X et al. [22] found that the pCR rate of grade 4 RIL vs. grade 0–3 RIL was 11.2% vs. 26.4% in EC patients, whilst Li Q et al. [19] observed that the pCR rate of grade 4 RIL vs. grade 0–3 RIL was 22.9% vs. 48.8% in EC patients. Our meta-analysis included three studies [19,22,34] and showed that the pCR rate was significantly lower in EC patients with severe RIL (OR = 0.44, 95% CI = 0.30–0.66, I^2^ = 0%; 3 studies with 819 samples; Table 2 and Figure 2).

In addition, Sherry A et al. reported that time-varying ALC was associated with the achievement of pCR (OR = 3.26, *p* = 0.0180) [33].

### 3.3. The Association between RIL and Survival Outcomes of EC Patients

Twelve studies [15,16,17,18,20,21,23,28,29,30,31,32] reported the association between RIL and the OS of EC patients. Our meta-analysis included five studies [15,16,17,20,23] and showed that severe RIL affected the OS of EC patients (HR = 1.50, 95% CI = 1.29–1.75, I^2^ = 6%; 5 studies with 1668 samples; Table 2 and Figure 2). Davuluri R et al. [21] (HR = 1.35, 95% CI = 1.02–1.78) and Xu C et al. [18] (HR = 1.284, 95% CI = 1.002–1.645) also observed that severe RIL affected the OS of EC patients. Nishida M et al. observed that grade 3–4 RIL after CRT was associated with worse OS of EC patients [29]. Cai S et al. showed that the OS significantly decreased in EC patients with grade 4 RIL (*p* < 0.001) [30]. Wang Q et al. found that for EC patients, the median OS of the lower ALC group (<0.8 × 10^9^/L) vs. higher ALC group (≥0.8 × 10^9^/L) was 35.5 vs. 49.8 months (*p* = 0.001) after propensity score matching [28]. Kroese T et al. observed that the median OS in EC patients with or without grade 4 RIL was 12.7 months and 42.5 months, respectively (*p* = 0.045) [31]. So T et al. showed that a higher lymphocyte nadir indicated lower risk for OS of EC patients (HR = 0.75 per 10^8^ cells/L, *p* = 0.003) by multivariable Cox regression model [32].

Six studies [16,17,19,20,21,30] reported the association between RIL and the PFS of EC patients. Our meta-analysis included four studies [17,19,20,35] and found that severe RIL affected the PFS of EC patients (HR = 1.51, 95% CI = 1.29–1.89, I^2^ = 52%; 4 studies with 1660 samples; Table 2 and Figure 2). In addition, Davuluri R et al. also observed that severe RIL affected the PFS of EC patients (HR = 1.58, 95% CI = 1.19–1.21) [21]. Cai S et al. found that the PFS of patients with grade 4 RIL was inferior to that of those with grade 0–3 RIL (3-year PFS: 20.7% vs. 45.4%, *p* = 0.048) [30].

Three studies [20,21,27] reported the association between RIL and the DMFS of EC patients. Deng W et al. [20] (HR = 1.27, 95% CI = 1.03–1.50) and Davuluri R et al. [21] (HR = 1.60, 95% CI = 1.13–2.27) observed that severe RIL affected the DMFS of EC patients. Routman D et al. [27] identified that, as a continuous variable, a higher lymphocyte nadir was significantly related to a decreased incidence of distant recurrence in EC patients (HR = 0.836, *p* = 0.0209) by multivariate analysis.

Three studies [20,21,27] reported the association between RIL and the LRFS of EC patients. Wang X et al. [17] (HR = 1.60, 95% CI = 1.17–2.17) and Deng W et al. [20] (HR = 1.23, 95% CI = 1.00–1.50) found that severe RIL was associated with inferior LRFS of EC patients, while Davuluri R et al. observed there was no association between RIL and the LRFS of EC patients (HR = 2.3, 95% CI = 0.80–1.90) [21].

In addition, Deng W et al. [20] (HR = 1.39, 95% CI = 1.10–1.76) and Davuluri R et al. [21] (HR = 1.60, 95% CI = 1.17–2.17) observed that severe RIL was associated with worse DSS of EC patients.

### 3.4. The Heterogeneity and Bias

There was relatively high heterogeneity among our meta-analyses in the association between RIL and the PFS of EC patients (I^2^ = 52%, Table 2). A sensitivity analysis was then performed, and we found that one study that might have caused heterogeneity [20]. With heterogeneity removed, the updated pooled meta-analysis still showed that severe RIL had a significant impact on EC patients’ PFS (HR = 1.70, 95% CI = 1.40–2.06, I^2^ = 0%; 3 studies with 905 samples; Appendix A).

The Egger’s test revealed no significant publication bias in our meta-analyses on the correlation of RIL and outcomes of EC patients (*p* > 0.05, Table 2).

### 3.5. Summary of Factors Associated with Severe RIL

Table 3 summarizes numerous dosimetric indicators and other risk factors linked to the development of RIL in each study based on multivariate analysis. Among dosimetric factors, larger PTV (four studies [16,17,22,32]), higher mean body dose (two studies [21,34]), higher effective dose to the immune cells (EDIC) (two studies [15,32]), and higher heart V10 (two studies [16,17]) were strong predictors of severe RIL. Higher total radiation dose [19], higher mean thoracic vertebrae dose [15], larger thoracic vertebrae V20 [15], larger body V10 [23], larger aorta V5 [23], larger heart V15 [23], larger T-spine V5 [23], and larger lung V10 [16] were also mentioned as being associated with developing severe RIL. In addition to dosimetric factors, other clinical factors associated with severe RIL were lower baseline ALC [16,17,32], greater tumor length [19,22] or higher clinical stage [17,34], distal EC [19,22], CRT [15] or chemotherapy type [21], older age [22], nonsmoking history [34], and lower ECOG performance status [16].

### 3.6. The Influence of CRT on ALC

As shown in Appendix A, nine studies [16,17,19,20,21,22,29,30,34] reported both the pretreatment ALC and the ALC nadir in EC patients who had CRT. The range of pretreatment ALC was 1400–1800 cells/μL, while the range of ALC nadir during or after CRT was 280–460 cells/μL, and the drop rate of ALC ranged from 73.99% to 83.33%. Five studies [16,17,19,20,22] reported the ALC after 4–8 weeks of the CRT completion, and the recovery of ALC ranged from 715 to 1200 cells/μL.

As shown in Figure 3a,b, seven studies recorded the weekly ALC from the initiation of radiotherapy [16,17,19,20,21,22,34]. All studies showed a significant drop of ALC at the first two weeks (ranging from 50.18% to 72.22%, Figure 3a), and the lowest ALC was identified during 4–6 weeks (Figure 3a and Appendix A). The pooled analysis showed that the mean median ALC dropped from 1615 cells/μL to 345 cells/μL at the 6th week (drop rate = 78.6%), and recovered to 976 cells/μL after 4–8 weeks from CRT completion.

## 4. Discussion

The current systematic review included 17 studies and used meta-analysis to show that the RIL of EC patients was associated with lower pCR rate and worse OS and PFS. We also evaluated a series of factors that might be associated with the incidence of RIL and summarized the ALC tendency of EC patients who received radiotherapy. Compared with the previous meta-analysis [13], which only included three cohorts and found a trend toward statistical significance that RIL was associated with inferior OS of EC patients, our systematic review was much more comprehensive, improving and expanding their findings.

Recently, immunotherapy has emerged as a standard treatment for many cancers, including EC [36,37]. Several immune checkpoint inhibitors that block the connection between PD-1 and PD-L1 are now being utilized to treat EC, such as nivolumab and pembrolizumab [38]. It should be emphasized that combining radiotherapy with immunotherapy is believed to be a promising strategy for improving therapeutic efficacy in a synergistic way [36]. Previous research has demonstrated that radiotherapy can boost tumor antigen presentation [39] as well as checkpoint inhibitor-induced antitumor immune responses in EC patients [40], which might improve the immunotherapy efficacy. However, radiotherapy also causes lymphopenia, which is associated with the reduction of the immune effector lymphocytes, poor response to immunotherapy, and worse prognosis [41]. Hence, it is crucial to lower the RIL rate and maintain an intact adaptive immune system during radiotherapy. Our study found that RIL was associated with worse outcomes of EC patients, which might be associated with reduced anti-cancer immune response and increased risk of infection [42]. Yin T et al. found that RIL was associated with the efficacy of immunotherapy. They found that radiotherapy increased the occurrence of lymphopenia (OR = 0.502, *p* = 0.035), and the overall response rate of EC patients without RIL was 29.4% while that of patients with RIL was 23.1%. EC patients receiving immunotherapy < 33.5 days (optimal cut-off value by ROC curves) after radiotherapy showed a poorer PFS compared to those ≥ 33.5 days (median PFS: 4.1 vs. 7.3 months, *p* = 0.008). Plenty of clinical trials of radiotherapy combined with immunotherapy for EC are now ongoing [5]. Our systematic review summarized the detailed ALC tendency among EC patients, which might provide helpful information for future clinical trials on the association between radiotherapy and immunotherapy in EC.

A series of clinical factors were associated with the incidence of RIL in EC. For the dosimetric parameters, the PTV volume, mean body dose, and heart V10 were mostly mentioned to be associated with RIL. The degree of RIL has been considered to depend on the dose/volume of blood flow as well as organs rich in lymphatics and lymphocytes in the radiation field [12]. The concept of EDIC was first introduced by Jin et al. at the 2017 ASTRO annual meeting, who considered the immune system as an organ at risk (OAR) and intended to examine the effects of radiotherapy dose on the immune system and the outcomes of lung cancer [43]. Studies observed that higher EDIC was associated with lower ALC and survival outcomes of small-cell lung cancer [44] and non-small-cell lung cancer [45,46]. Similar EDIC models were applied to EC, as it is also within the thoracic cavity. Liu M et al. found that an EDIC higher than 7.11 Gy predicted a lower ALC and lower OS in EC [15]. Xu C et al. showed that EDIC maintained a risk factor for severe RIL, after controlling for other risk factors such as age, ECOG, and PTV. Furthermore, a higher EDIC was linked to poorer OS, PFS, and DMFS [18]. So T et al. observed that EDIC had a significantly negative correlation with ALC nadir, and that a higher EDIC predicted worse OS (<2 Gy, >2 and <4 Gy, and ≥4 Gy groups) [32]. In these studies, various formulae were designed to estimate EDIC based on the doses to the lung, heart, and liver (which are the organs with the largest pool of circulating or resident immune cells) and the remaining body [15,18,44,46]. However, there might be more organs to account for, such as the spleen. A study showed that higher splenic dose increases the risk of lymphopenia in a series of upper abdominal cancers, including EC [47,48]. Saito et al. showed that when the mean splenic dose increased by 1 Gy, the predicted ALC decreased by 2.9% in EC patients [49]. Our systematic review also showed that a higher dose to the thoracic vertebrae or aorta was associated with the incidence of RIL in EC. Further improvements to the EDIC model of EC with more specified organs are required. EDIC should be introduced in the design of the radiotherapy plan for EC, which might be an important countermeasure to lower the incidence of RIL and improve the prognosis of EC patients.

In addition to dosimetric factors, there were also other factors associated with severe RIL, such as lower baseline ALC, greater tumor length, higher clinical stage, distal EC, CRT, chemotherapy regime, older age, no smoking at diagnosis, and lower ECOG performance status. Greater tumor length and higher clinical stage were associated with larger irradiated area. Distal esophageal location spans across the heart and close to the spleen, the dose of which might have a higher rate of causing RIL. Consistently, our systematic review showed that the rate of grade 4 RIL was higher in the studies [18,20,21,23,34] with majority lower EC (38.9–50.0%) than those [16,19,22,30] with majority upper/middle EC (21.8–31%) (Table 1). The histology of EC differs greatly between the United States and Eastern Asia (Table 1); the United States had more EAC whilst Eastern Asia had more ESCC [50]. Our results show that there is a much higher rate of grade 4 RIL in the United States than in Eastern Asia (Table 1). However, this might be a result of the location difference rather than a histology difference, as the radiotherapy plan does not differ much between ESCC and EAC. The radiotherapy technique might also influence the RIL rate. Lin et al. conducted a randomized phase IIB trial and showed that the rates of grade 4 RIL were 52% and 27% in locally advanced EC patients under IMRT and PBT, respectively [51]. They found that PBT resulted in considerably reduced doses to total lung indicators (V5, 41.4% vs. 19.7%; V20, 13.6% vs. 8.4%; mean lung dose, 8.4 vs. 4.8 Gy; all *p* < 0.001) and also mean doses to the liver (12.1 vs. 2.4 Gy; *p* < 0.001) and heart (19.8 vs. 11.3 Gy; *p* < 0.001) [51], which might greatly lower the EDIC. Davuluri R et al. also found that the incidence of grade 4 RIL is significantly reduced in EC patients treated with proton beam therapy (PBT) when compared to intensity-modulated radiation therapy (IMRT) (15.5% vs. 33.1%, *p* < 0.001) [21].

There are some limitations in our systematic review. First, since the included studies were all retrospectively designed, there were many potential confounders in linking RIL and clinical outcomes in the included studies. For example, larger PTV, longer tumor length, or GEJ location were more associated with RIL. However, they might also be associated with more advanced disease, which also has worse outcomes. Despite using the multivariate results in the meta-analysis, future data from prognostic and randomized designed studies are required. Second, the details of radiotherapy conduction were often missing, such as the fields and organs at risk. Third, there was heterogeneity in the definition of severe RIL. The method of pooling data from studies with different definitions of severe RIL was from a previous study on the association between RIL and the prognosis of lung cancer [52]. However, there was relatively low heterogeneity in our meta-analysis of the OS (I^2^ = 6%) and PFS after sensitivity analysis (I^2^ = 0%). Fourth, our included studies showed the association between RIL and the outcomes of EC patients; however, whether the RIL could influence the benefits of immunotherapy was not studied among these cohorts. Moreover, the rate of RIL in our included studies was mostly recorded during radiotherapy; whether the lymphopenia could recover after treatment and whether prolonged treatment-related lymphopenia could impact the benefits of immunotherapy was meaningful, as much of the data on the use of immunotherapy for locally advanced EC is in the adjuvant setting, such as in the Checkmate 577 trial or the ongoing SKYSCRAPER 07 trial. Although we found that the mean median ALC recovered to 976 cells/μL after 4–8 weeks from CRT completion, it was still much lower than the initial ALC, which was 1615 cells/μL.

## 5. Conclusions

Our systematic review showed that severe RIL might be associated with a lower pCR rate, worse OS, and worse PFS of EC patients. The ALC nadir was observed during 4–6 weeks after the start of CRT, with a significant recovery during 4–8 weeks after CRT completion. The dosimetric factors associated with severe RIL included larger PTV, higher mean body dose, higher EDIC, and higher heart V10. An improved EDIC model is required to quantify the dose to lymphocytes with more accuracy, and EDIC should be introduced in the design of radiotherapy plans in EC. Minimizing the dosimetric factors, especially among patients with lower baseline ALC, greater tumor length, higher clinical stage, and distal EC, might potentially lower the rate of severe RIL and improve the outcomes of EC.

## Figures and Tables

**Figure 1 cancers-14-03024-f001:**
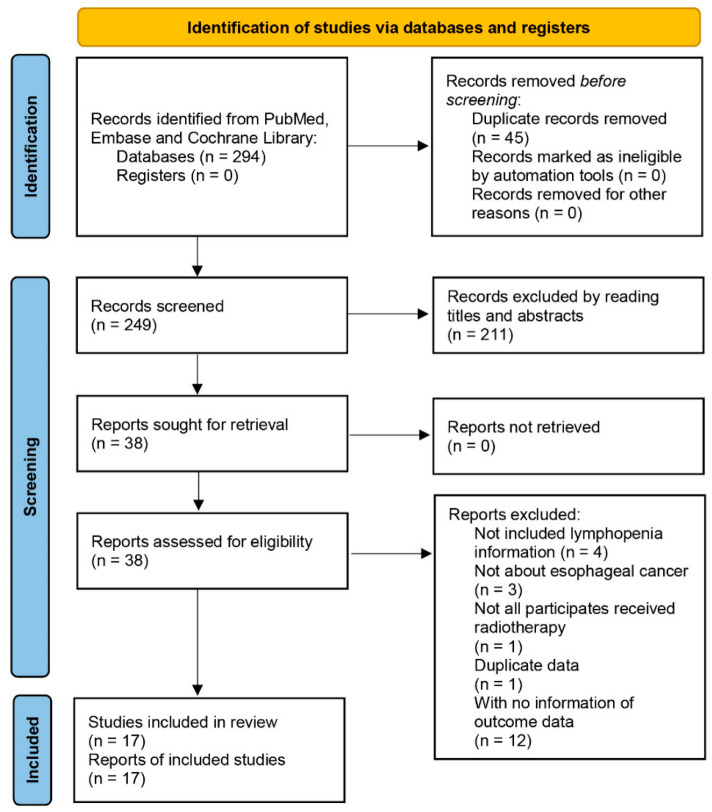
Flowchart of selection process in this systematic review. This is the PRISMA 2020 flow diagram for new systematic reviews, which only includes searches of databases and registers.

**Figure 2 cancers-14-03024-f002:**
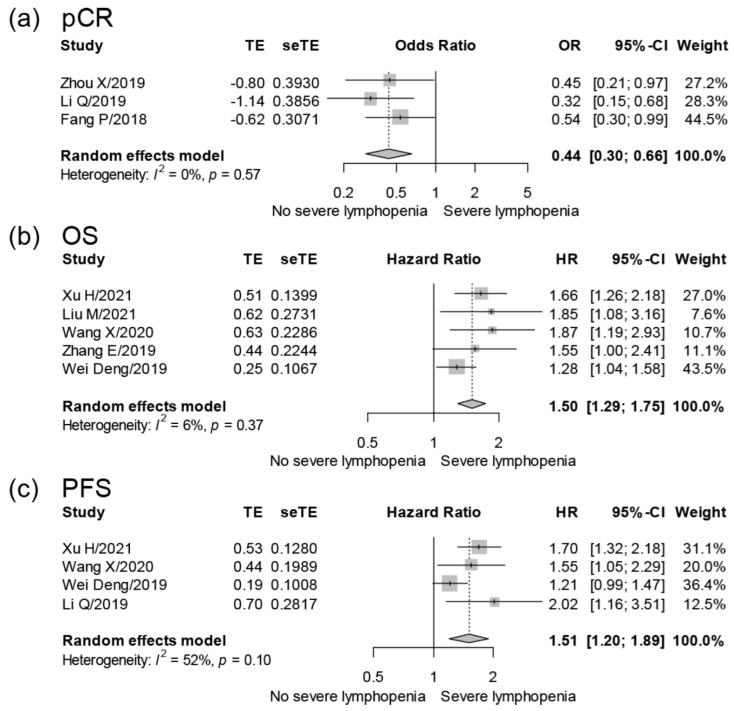
Forest plots depicting meta-analyses for the association between severe RIL and the outcomes of EC patients; meta-analyses for the correlation of severe RIL and pCR (**a**), OS (**b**), and PFS (**c**) of EC patients. The big diamond at the bottom of the plot symbolizes the pooled OR or HR of all studies. The diamond’s width corresponds to 95% CI. For these included multivariate analyses, Zhou X et al. included variables of lymphopenia, tumor length, and baseline ALC. Li Q et al. included variables of lymphopenia, pretreatment platelets, chemotherapy, and interval to surgery in the meta-analysis of pCR, and included age, sex, primary tumor length, clinical stage, chemotherapy, interval to surgery, and pCR status in the meta-analysis of PFS. Fang P et al. included variables of lymphopenia, age, current smoker, clinical stage, histology, differentiation, tumor length and location, radiotherapy dose and modality, induction chemotherapy, and chemotherapy regimen. Xu H et al. included variables of lymphopenia, sex, tumor location and length, induction chemotherapy, clinical stage, radiotherapy modality, and baseline albumin and hemoglobin. Liu M et al. included variables of lymphopenia; age; sex; ECOG score; tumor location; clinical stage; chemotherapy; prescribed radiotherapy dose; pretreatment NLR; mean TVB dose; TVB V5, V10, and V20; and EDIC. Wang X et al. included variables of lymphopenia, age, sex, smoking, drinking, tumor length and location, clinical stage, treatments (concurrent chemotherapy and chemotherapy regimen), radiotherapy technology and dose, and baseline ALC. Zhang E et al. included variables of lymphopenia, clinical stage, and surgery. Deng W et al. included variables of lymphopenia, baseline ALC, sex, age, ECOG score, tumor length and location, histology, differentiation, clinical stage, and treatments (induction and concurrent chemotherapy and surgery).

**Figure 3 cancers-14-03024-f003:**
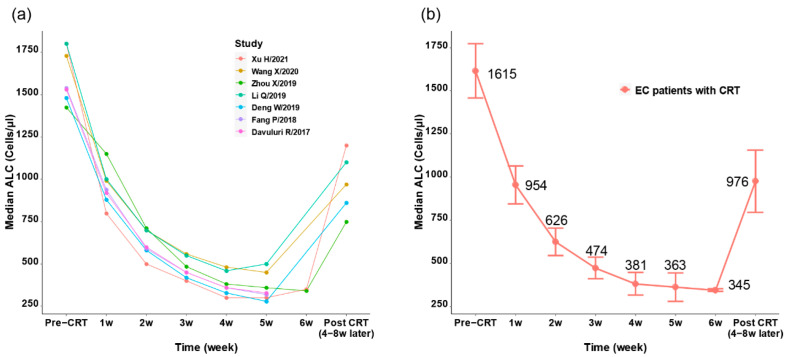
The tendency of ALC in EC patients who received CRT. (**a**) Serial median ALC in EC patients who received CRT from various studies. (**b**) Merged mean ALC in EC patients who received CRT; the circle represents the mean median ALC of studies and the error bars represent the standard deviation.

**Table 1 cancers-14-03024-t001:** The characteristics of each included study.

First Author/Year	Ref.	Region	No. Patients/Stage	Age (Years)	Histology	Tumor Location	Median Follow-Up (Months)	RT Technique	Dose (Gy)	Chemotherapy	Surgery Rate after CRT	Definition of Severe RIL/Data Collection Time ^a^	Severe RIL Rate
**For meta-analysis**													
Xu H/2021	[16]	China	436/I-IVA	Median (range) = 59 (27–74)	ESCC	Upper/Middle (86.7%), Lower (13.3%)	21.7	IMRT (73.6%), 3D-CRT (26.4%)	Median (range) = 60 (40–70), with 25–30 fractions.	Concurrent weekly chemotherapy, with 54.8% cisplatin + taxane, 17.2% cisplatin + 5-FU, 28% other	0%	G4/During RT	23.6%
Liu M/2021	[15]	China	99/II-IVA	Median (range) = 67 (43–83)	ESCC	Upper/Middle (75%), Lower (25%)	24.7	IMRT (100%)	Median (range) = 55.75 (46–66), with 1.8–2.2 per fraction	Concurrent chemoradiotherapy (59%), sequential chemoradiotherapy (9%); the others were RT alone	0%	<300/Within 2 months after RT started	NA
Wang X/2020	[17]	China	189/I-IVA	Median (range) = 67 (44–92)	ESCC	Upper/Middle (76.2%), Lower (23.8%)	46	IMRT (65.1%), 3D-CRT (34.9%)	50–68	Concurrent chemotherapy receiving platinum and 5-FU doublet chemotherapy (16.9%), followed by combination of platinum and taxane (12.7%)	0%	<380/During RT	58.2%
Zhou X/2019	[22]	China	286/II-IVA	Median (range) = 67 (47–84)	ESCC	Upper/Middle (88.1%), Lower (11.9%)	NA	IMRT (76.6%), Proton (23.4%)	Median = 50.4, with 1.8 per fraction	Concurrent cisplatin + docetaxel (72.4%) or S-1 (27.6%)	NA	G4/During RT	31.0%
Zhang E/2019	[23]	United States	189/ I-III	Median (range) = 65 (35–84)	EAC (78%), other (22%)	Upper/Middle (15%), Lower (85%)	27.6	NA	Median (range) = 50.4 (41.4–70.2)	Concurrent carboplatin/paclitaxel (55%) or 5-FU (40%)-based regimens	68%	G4/During RT	45.0%
Li Q/2019	[19]	China	220/II-III	Median (range) = 56 (42–73)	ESCC	Upper/Middle (77.3%), Lower (22.7%)	24.8	IMRT (33.2%), 3D-CRT (66.8%)	Median (range) =40 (40–50.4), with 20–25 fractions	Concurrent cisplatin + vinorelbine (57.3%), cisplatin + taxane (29.5%), or cisplatin + 5-FU (13.2%)	100%	G4/During RT	21.8%
Deng W/2019	[20]	United States	755/I-III	Median (IQR) = 64 (57–71)	ESCC (17.6%), EAC (81.8%), other (0.5%)	Upper/Middle (14.8%), Lower (85.2%)	65.5	IMRT (67.7%), Proton (32.3%)	Median (range) = 50.4 (41.4–66.0)	Concurrent weekly taxane, 5-FU, or platinum-based compound	49.9%	G4/During RT	38.9%
Fang P/2018	[34]	United States	313/I-IVA	Mean (SD) = 59.3 (10.8)	ESCC (5%), EAC (95%)	Upper/Middle (2.3%), Lower (97.7%)	NA	IMRT (67%), Proton (33%)	Mean (SD) = 50 (1.9)	Concurrent taxane + 5-FU (43.1%), platinum + 5-FU (38.7%), platinum + taxane (10.9%), or other regimens (7.4%)	100%	<350/During RT	56.0%
**Only for systematic review**													
Wang Q/2021	[28]	China	476/I-IV	Median (range) = 63 (37–85)	ESCC	Upper/Middle (84.2%), Lower (15.8%)	Total 60 months	IMRT (65.8%), 3D-CRT (34.2%)	50–60, with 25–33 fractions	Concurrent vs. Non-concurrent = 12.2% vs. 87.8%, Paclitaxel vs. 5-FU = 58% vs. 42%	NA	<800/During RT	54.2%
Nishida M/2021	[29]	Japan	298	NA	EC	NA	NA	NA	NA	CRT	NA	G3-4/During RT	NA
Kroese T/2021	[31]	Netherlands	219/I-IV	NA	EC	Upper/Middle (30.6%), Lower (69.4%)	26.7	IMRT	41.4 or 50.4, with 1.8 per fraction	Concurrent chemotherapy for all, CROSS regime	100%	G4/During RT	16.0%
Cai S/2021	[30]	China	146/I-IVA	Median (range) = 71 (50–91)	ESCC (95.2%), EAC and others (4.8%)	Upper/Middle (83.6%), Distal (16.4%)	17.9	IMRT	45–50.4 for neoadjuvant CRT and 60–64 for definitive RT, with 1.8–2 per fraction	15.1% received neoadjuvant chemotherapy, 45.2% received consolidation chemotherapy; Carboplatin and paclitaxel were most used	21.90%	G4/During RT and 1 week after RT	24.7%
Xu C/2020	[18]	United States	488/I-IV	Median (range) = 61 (20–84)	ESCC (9.8%), EAC (90.2%);	Upper/Middle (7%), Lower (93%)	29.6	IMRT	45–50.4	Concurrent chemotherapy for all	55.9%	G4/During RT	50.0%
So T/2020	[32]	Hong Kong	92/II-IVA	<65 year (45.7%), ≥65 (54.3%)	ESCC	NA	16.9	IMRT (43.5%), 3D-CRT (56.5%)	41.4, with 23 fractions	Concurrent weekly carboplatin, with area under the curve = 2 and paclitaxel = 50 mg/m^2^	100%	G3-4/During RT and after 2 months after completion of RT	NA
Sherry A/2019	[33]	United States	93/II-III	Median (IQR) = 64 (55–72)	ESCC (17%), EAC (82%)	Upper/Middle (12%), Lower (88%)	19.2	IMRT (35%), 3D-CRT (65%)	50.4 (IQR, 50–50.4), with 28 fractions (IQR, 25–28)	Concurrent platinum + taxane (78%), platinum + 5-FU (14%), other (7%)	71%	NA/During RT	NA
Routman D/2017	[27]	United States	176/NA	NA	EC	NA	39.6	NA	NA	NA	NA	G4/During RT and before surgery	46.0%
Davuluri R/2017	[21]	United States	504/I-III	Median (SD) = 62.5 (11.2)	ESCC (15%), EAC (85%)	Upper/Middle (11%), Lower (89%)	32.1	IMRT (63%), Proton (37%)	50.4 for all	Concurrent taxane + 5-FU (49%), platinum + taxane (13%), platinum + 5-FU (31%), other (7%)	46%	G4/During RT	27.0%

Abbreviations: NA—not available; IQR—interquartile range; SD—standard deviation; ESCC—esophageal squamous cell carcinoma; EC—esophageal carcinoma; EAC—esophageal adenocarcinoma; IMRT—Intensity-Modulated Radiation Therapy; 3D-CRT—three-dimensional conformal radiation therapy; CRT—chemoradiation therapy; 5-FU—5-fluorouracil; CROSS—chemoradiotherapy for esophageal cancer followed by surgery study; RIL—radiation-induced lymphopenia; RT—radiotherapy. ^a^ G3/4—grade 3/4 lymphopenia according to the standardized Common Terminology Criteria for Adverse Events (CTCAE). The unit used for the definition of RIL is cells/μL.

**Table 2 cancers-14-03024-t002:** The meta-analyses of associations between RIL and the outcomes of EC patients.

Group	Number of Studies	Sample Size	Pooled OR or HR (95% CI)	I^2^ (%)	Egger’s *p* Value
pCR	3	819	0.44 (0.30–0.66) ^a^	0	0.4391
OS	5	1668	1.50 (1.29–1.75) ^b^	6	0.1053
PFS	4	1660	1.51 (1.20–1.89) ^b^	52	0.2824

Abbreviations: pCR—pathologic complete response; OS—overall survival; PFS—progression-free survival; OR—odds, ratio; CI—confidence interval; HR—hazard ratio. ^a^ OR (95% CI); ^b^ HR (95% CI).

**Table 3 cancers-14-03024-t003:** Risk factors for developing severe RIL in EC patients, as determined via multivariate analysis.

First Author/Year	Ref.	Dosimetric Factors ^a^^b^	Others ^a^	Non-Significant Factors ^b^
Xu H/2021	[16]	**Larger PTV, higher heart V10**, higher lung V10	**Lower baseline ALC**, lower ECOG performance status	Sex, smoking history, alcohol history, tumor location (upper vs. middle/distal), clinical N stage (N0–1 vs. N2–3), lung V5, lung V20, MLD, MHD, VB V10, mean VB dose
Liu M/2021	[15]	**Higher EDIC**, higher mean dose, higher V20 of TVB	Chemotherapy regimen (CRT vs. RT alone)	V5 and V10 of TVB, RT dose (<60 vs. ≥60 Gy)
Wang X/2020	[17]	**Larger PTV, higher heart V10**	**Higher clinical stage, lower baseline ALC**	Tumor length, lung V5, lung V10, lung V20, lung V30, lung V40, lung mean dose, heart V5, heart V30, heart V40, heart mean dose
So T/2020	[32]	**Larger PTV, higher EDIC**	**Lower baseline ALC**	The number of courses of chemotherapy (5 courses vs. fewer than 5 courses)
Zhou X/2019	[22]	**Larger PTV**	**Distal EC, tumor length > 5 cm**, older age	MLD
Zhang E/2019	[23]	Heart V15 > 73%, TVB V5 > 72%, body V10 > 18%, aorta V5 > 93%	Lower baseline TLC	Total lung V5 > 50%, spleen V20 > 45%
Li Q/2019	[19]	Higher radiation dose (>40 Gy)	**Distal EC, tumor length > 5 cm**	Age, alcohol history, dose per fraction > 2.0 Gy, IMRT vs. 3D-CRT, clinical TNM stage (II vs. III), pretreatment ALC (<1.8 vs. ≥1.8 × 10^9^/L)
Fang P/2018	[34]	**Higher mean body dose**	**Higher clinical stage**, no smoking at diagnosis	Age, tumor histology, tumor differentiation, tumor location, tumor size, RT modality, induction chemotherapy or chemotherapy type between the groups
Davuluri R/2017	[21]	**Higher mean body dose**	Concurrent Taxane/5-FU vs. platinum/5-FU	Age, comorbidities, tumor characteristics (location, length, stage, histology, differentiation), surgery, RT modality, induction chemotherapy

Abbreviations: PTV—planning target volume; TVB—thoracic vertebrae; ECOG—Eastern Cooperative Oncology Group; CRT—chemoradiation therapy; RT—radiotherapy; ALC—absolute lymphocyte count; TLC—total lymphocyte count; Distal EC—distal esophageal cancer; 5-FU—5-fluorouracil; IMRT—intensity-modulated radiation therapy; 3D-CRT—three-dimensional conformal radiation therapy; MLD—mean lung dose; MHD—mean heart dose; VB—vertebral body. ^a^ Factors with 2 or more significant results are in bold; ^b^ Vx—percentage of the total lung, heart, or VB volume receiving more than x Gy.

## Data Availability

Not applicable.

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
