# Peer review of "Severe Radiation-Induced Lymphopenia Affects the Outcomes of Esophageal Cancer: A Comprehensive Systematic Review and Meta-Analysis"

_cancers, 2022, doi:10.3390/cancers14123024_

Round 1

Reviewer 1 Report

This systematic review and meta analysis was conducted based on established methodologies, with two independent reviewers screening studies, with the possibility of asking a third reviewer to resolve any conflicts in study retention for the final analysis. Given the paucity of data in the literature, I congratulate the authors on pooling studies of moderate to high quality in trying to establish an association between RIL and oncologic outcomes. Due to the retrospective nature of many references and the lack of access to individual patient data, I would rephrase the conclusion with softer terms than the current manuscript version.

Author Response

Point 1: This systematic review and meta analysis was conducted based on established methodologies, with two independent reviewers screening studies, with the possibility of asking a third reviewer to resolve any conflicts in study retention for the final analysis. Given the paucity of data in the literature, I congratulate the authors on pooling studies of moderate to high quality in trying to establish an association between RIL and oncologic outcomes. Due to the retrospective nature of many references and the lack of access to individual patient data, I would rephrase the conclusion with softer terms than the current manuscript version.

Response 1: Thank you for your kind comments. Indeed, our meta-analysis only included retrospective studies and lacked access to individual patient data. We have already described this as the first limitation of our study in the discussion part. Following your suggestion, we have now revised the words in the conclusion part of our study, and we used softer terms like “might” to replace the previous manuscript version. Besides, we have now carefully checked our manuscript and revised the potential grammar mistakes. We hope our revision could meet your concerns. Thank you again for your comments.

Reviewer 2 Report

This report is important, but there are some points to revise before publishing.

1. I do not find your register in PROSPERO when I search following words “Radiation Induced Lymphopenia”. Did you register?

2. I think not only radiotherapy but also chemotherapy plays an important role in treatment-related lymphopenia. So, “Radiotherapy plays a primary role” is overstatement.

3. What do authors think about countermeasure ,for example proton beam therapy, for “Radiation Induced Lymphopenia”?

Author Response

This report is important, but there are some points to revise before publishing.

Point 1: I do not find your register in PROSPERO when I search following words “Radiation Induced Lymphopenia”. Did you register?

Response 1: We did not register in PROSPERO. However, we registered at Open science (https://osf.io/prereg/) as the editor of Cancers suggested, and the registration number of our systematic review is 10.17605/OSF.IO/HK7QU.

Point 2: I think not only radiotherapy but also chemotherapy plays an important role in treatment-related lymphopenia. So, “Radiotherapy plays a primary role” is overstatement.

Response 2: Thank you for your kind suggestion. Indeed, the chemotherapy also causes the treatment-related lymphopenia. However, radiotherapy might play a more important role in treatment-related lymphopenia. A study revealed that total lymphocyte counts did not change following neoadjuvant chemotherapy in non-small-cell lung cancer, but steep declines were noted after the initiation of thoracic radiotherapy [1]. There were also studies showed that although both radiotherapy and concurrent CRT significantly decreased the mean absolute number of lymphocytes compared to pretreatment levels, no differences were detected in the characteristics of the lymphopenia induced by those two treatments in cervical cancer [2,3], which implied that radiotherapy might play a more important role in the incidence of lymphopenia. We have now changed the sentence to “Radiotherapy may play a primary role in the incidence of treatment-related lymphopenia in cancer patients” to avoid overstatement. We also added new sentences to describe the related issue as follows:

“Radiotherapy may play a primary role in the incidence of treatment-related lymphopenia in cancer patients [4], since lymphocytes are so radiosensitive that even a fraction of 2 Gy can kill half of the irradiated lymphocytes [5]. The radiation induced lymphopenia (RIL) affects all lymphocyte subsets (CD3+, CD4+, CD8+ T cells, B cells, and natural killer cells) [6]. A study revealed that total lymphocyte counts did not change after neoadjuvant chemotherapy in non-small cell lung cancer, while steep declines were found after the initiation of thoracic radiotherapy [1]. There were also studies showed that although both radiotherapy and concurrent CRT significantly decreased the mean absolute number of lymphocytes compared to pretreatment levels, no differences were detected in the characteristics of the lymphopenia induced by the two treatments in cervical cancer [2,3].”

Point 3: What do authors think about countermeasure, for example proton beam therapy, for “Radiation Induced Lymphopenia”?

Response 3: Thank you for your valuable comments. We think the key part of the countermeasure is to lower the effective dose to the immune cells (EDIC). As we described in the discussion part, the concept of EDIC was first introduced by Jin et al. at the 2017 ASTRO annual meeting, who considered the immune system as an organ at risk (OAR). Studies observed that the higher EDIC was associated with lower ALC and survival outcomes of small cell lung cancer [7], non-small cell lung cancer [8,9], and esophageal cancer [10-12]. In these studies, various formulae were designed to estimate EDIC based on the doses to the lung, heart, liver (which contains the organs with the largest pool of circulating or resident immune cells) and the remaining body [7,9-11]. However, further improvements to the EDIC model of EC with more specified organs, such as spleen, are required. An improved EDIC should be introduced in the design of the radiotherapy plan for EC to lower the “Radiation Induced Lymphopenia”. As you mentioned, the use of proton beam therapy is certainly a good strategy to lower the EDIC.

We have now added new sentences to show that lowering the EDIC is an important countermeasure to lower the RIL and improve the prognosis of EC patients.

“Further improvements to the EDIC model of EC with more specified organs are required. The EDIC should be introduced in the design of the radiotherapy plan for EC, which might be an important countermeasure to lower the RIL and improve the prognosis of EC patients.”

In addition, we have now carefully checked our manuscript and revised the potential grammar mistakes. We hope our revision could meet your concerns. Thank you again for your valuable comments.

References

  1. Campian, J.L.; Ye, X.; Brock, M.; Grossman, S.A. Treatment-related lymphopenia in patients with stage III non-small-cell lung cancer. Cancer Invest 2013, 31, 183-188, doi:10.3109/07357907.2013.767342.
  2. van Meir, H.; Nout, R.A.; Welters, M.J.; Loof, N.M.; de Kam, M.L.; van Ham, J.J.; Samuels, S.; Kenter, G.G.; Cohen, A.F.; Melief, C.J.; et al. Impact of (chemo)radiotherapy on immune cell composition and function in cervical cancer patients. Oncoimmunology 2017, 6, e1267095, doi:10.1080/2162402X.2016.1267095.
  3. Santin, A.D.; Hermonat, P.L.; Ravaggi, A.; Bellone, S.; Roman, J.; Pecorelli, S.; Cannon, M.; Parham, G.P. Effects of concurrent cisplatinum administration during radiotherapy vs. radiotherapy alone on the immune function of patients with cancer of the uterine cervix. Int J Radiat Oncol Biol Phys 2000, 48, 997-1006, doi:10.1016/s0360-3016(00)00769-0.
  4. Wang, X.; Wang, P.; Zhao, Z.; Mao, Q.; Yu, J.; Li, M. A review of radiation-induced lymphopenia in patients with esophageal cancer: an immunological perspective for radiotherapy. Ther Adv Med Oncol 2020, 12, 1758835920926822, doi:10.1177/1758835920926822.
  5. Yovino, S.; Kleinberg, L.; Grossman, S.A.; Narayanan, M.; Ford, E. The etiology of treatment-related lymphopenia in patients with malignant gliomas: modeling radiation dose to circulating lymphocytes explains clinical observations and suggests methods of modifying the impact of radiation on immune cells. Cancer Invest 2013, 31, 140-144, doi:10.3109/07357907.2012.762780.
  6. Domouchtsidou, A.; Barsegian, V.; Mueller, S.P.; Best, J.; Ertle, J.; Bedreli, S.; Horn, P.A.; Bockisch, A.; Lindemann, M. Impaired lymphocyte function in patients with hepatic malignancies after selective internal radiotherapy. Cancer Immunol Immunother 2018, 67, 843-853, doi:10.1007/s00262-018-2141-0.
  7. Yu, Y.; Fu, P.; Jin, J.Y.; Gao, S.; Wang, W.; Machtay, M.; Wang, L.; Kong, F.S.; Yu, J. Impact of effective dose to immune cells (EDIC) on lymphocyte nadir and survival in limited-stage SCLC. Radiother Oncol 2021, 162, 26-33, doi:10.1016/j.radonc.2021.06.020.
  8. Jin, J.Y.; Hu, C.; Xiao, Y.; Zhang, H.; Paulus, R.; Ellsworth, S.G.; Schild, S.E.; Bogart, J.A.; Dobelbower, M.C.; Kavadi, V.S.; et al. Higher Radiation Dose to the Immune Cells Correlates with Worse Tumor Control and Overall Survival in Patients with Stage III NSCLC: A Secondary Analysis of RTOG0617. Cancers (Basel) 2021, 13, doi:10.3390/cancers13246193.
  9. Ladbury, C.J.; Rusthoven, C.G.; Camidge, D.R.; Kavanagh, B.D.; Nath, S.K. Impact of Radiation Dose to the Host Immune System on Tumor Control and Survival for Stage III Non-Small Cell Lung Cancer Treated with Definitive Radiation Therapy. Int J Radiat Oncol Biol Phys 2019, 105, 346-355, doi:10.1016/j.ijrobp.2019.05.064.
  10. Liu, M.; Li, X.; Cheng, H.; Wang, Y.; Tian, Y. The Impact of Lymphopenia and Dosimetric Parameters on Overall Survival of Esophageal Cancer Patients Treated with Definitive Radiotherapy. Cancer Manag Res 2021, 13, 2917-2924, doi:10.2147/CMAR.S297010.
  11. Xu, C.; Jin, J.Y.; Zhang, M.; Liu, A.; Wang, J.; Mohan, R.; Kong, F.S.; Lin, S.H. The impact of the effective dose to immune cells on lymphopenia and survival of esophageal cancer after chemoradiotherapy. Radiother Oncol 2020, 146, 180-186, doi:10.1016/j.radonc.2020.02.015.
  12. So, T.H.; Chan, S.K.; Chan, W.L.; Choi, H.; Chiang, C.L.; Lee, V.; Lam, T.C.; Wong, I.; Law, S.; Kwong, D.; et al. Lymphopenia and Radiation Dose to Circulating Lymphocytes With Neoadjuvant Chemoradiation in Esophageal Squamous Cell Carcinoma. Adv Radiat Oncol 2020, 5, 880-888, doi:10.1016/j.adro.2020.03.021.

Reviewer 3 Report

Summary:

This systematic review evaluates the outcome of patients with esophageal cancer who developed severe radiation-induced lymphopenia after radiotherapy. Seventeen studies were included, and a meta-analysis was performed on eight studies in this systematic review. The study found that esophageal cancer patients developing radiation-induced lymphopenia after radiotherapy had a lower pathological complete response rate, poor overall survival, and progression-free survival.

Strength: The systematic review provides the meta-analysis specific to esophageal cancer, evaluating the association between the radiation-induced lymphopenia and prognostic factors, including the complete pathological response (pCR), overall survival (OS), and progression-free survival (PCS).    

Limitations:

-        Authors must define the question. What review question do the authors aim to answer with this meta-analysis. The systematic review must emphasize the clinical value of the results and findings from this meta-analysis.

-        Authors must restate how this systematic review and meta-analysis has the advantage over previous systematic review cited as [10]

Abstract: It would be appropriate to organize the Abstract and Summary in sections including Purpose, Methods and Materials, Results, and Conclusions.

1.      In the Introduction section, paragraph four, the authors have stated that “Previous meta-analysis showed there was no significant association between RIL and the OS of EC patients. To be noted, this study only analyzed the association between the RIL and the OS of EC patients, and we found there were some related studies that did not included in this meta-analysis. Therefore, we performed this more comprehensive systematic review and meta-analysis to explore the association between RIL and the outcomes of EC patients.”

The cited systematic review showed a significant association between the RIL and OS. Authors are advised to revisit the cited meta-analysis [10] that included three esophageal cancer studies and suggested that the adjusted hazard ratio (HR) for overall survival (OS) was consistently higher than one and trended towards statistical significance.

2.      In the discussion section, paragraph 1 authors state that “Compared with the previous meta-analysis which only included 3 cohorts and found no significant association between RIL and the OS of EC patients, our systematic review was much more comprehensive”.

Please restate this statement after revision of your interpretations from the cited meta-analysis.   

Specific comments:

·        A space is required between the end of the statement and the bracket, including the citation numbers.

·        Authors often used the word “would” in the text. For example, “The meta-analysis would be conducted with equal or over 3 individual results……”. Authors are advised to use appropriate tense or must take help from professional writers. 

Author Response

Point 1: Authors must define the question. What review question do the authors aim to answer with this meta-analysis. The systematic review must emphasize the clinical value of the results and findings from this meta-analysis.

Response 1: Thank you for your kind suggestion. We have now added new sentences at the end of the introduction section to describe our review question as follows:

Our systematic review seeks to determine whether RIL is associated with poorer outcomes in EC patients and to summarize the risk factors for severe RIL in EC patients. The results of our study might provide valuable information for radiotherapy strategy in EC patients.

Point 2: Authors must restate how this systematic review and meta-analysis has the advantage over previous systematic review cited as [10]

Response 2: Thank you for your comments. We have now revised the introduction and discussion part related to the previous systematic review.

In the introduction section, we have added new sentences to describe the reason why we perform this systematic review over the previous one as follows:

“There was a previous meta-analysis that found RIL was associated with inferior OS of severe solid tumors [1]. This meta-analysis included 3 EC studies [2-4] and found the individual adjusted hazard ratio (HR) estimated in these studies was consistently larger than 1, and pooled HR demonstrated a trend toward statistical significance (HR = 1.46, 95% confidence interval [95%CI] = 0.95-2.23). To be noted, this study only analyzed the association between the RIL and the OS of EC patients, and we found there were some related studies that were not included in this meta-analysis [5,6]. Therefore, we performed this more comprehensive systematic review and meta-analysis. Our systematic review seeks to determine whether RIL is associated with poorer outcomes in EC patients and to summarize the risk factors for severe RIL in EC patients. The results of our study might provide valuable information for radiotherapy strategy in EC patients.

In the first paragraph of the discussion section, we have summarized our findings and shown that our meta-analysis has advantages over previous systematic review as follows:

“The current systematic review included 17 studies and used meta-analysis to show that the RIL of EC patients was associated with lower pCR rate and worse OS and PFS. We also exhibited a series of factors that might be associated with the incidence of RIL and summarized the ALC tendency of EC patients who received radiotherapy. Compared with the previous meta-analysis [1] which only included 3 cohorts and found a trend toward statistical significance that RIL was associated with inferior OS of EC patients, our systematic review was much more comprehensive, which improved and expanded their findings.

Point 3: Abstract: It would be appropriate to organize the Abstract and Summary in sections including Purpose, Methods and Materials, Results, and Conclusions.

Response 3: Thank you for your comments. However, the instructions for authors of Cancers demand that the abstract should be a single paragraph without headings. Therefore, we could not organize the abstract into subsections.

Point 4: In the Introduction section, paragraph four, the authors have stated that “Previous meta-analysis showed there was no significant association between RIL and the OS of EC patients. To be noted, this study only analyzed the association between the RIL and the OS of EC patients, and we found there were some related studies that did not included in this meta-analysis. Therefore, we performed this more comprehensive systematic review and meta-analysis to explore the association between RIL and the outcomes of EC patients.”

The cited systematic review showed a significant association between the RIL and OS. Authors are advised to revisit the cited meta-analysis [10] that included three esophageal cancer studies and suggested that the adjusted hazard ratio (HR) for overall survival (OS) was consistently higher than one and trended towards statistical significance.

Response 4: Thank you for your kind advise. We have now added new sentences with more details of this previous meta-analysis in this paragraph as follows:

“There was a previous meta-analysis that found RIL was associated with inferior OS of severe solid tumors [1]. This meta-analysis included 3 EC studies [2-4] and found the individual adjusted hazard ratio (HR) estimated in these studies was consistently larger than 1, and pooled HR demonstrated a trend toward statistical significance (HR = 1.46, 95%confidence interval [95%CI] = 0.95-2.23). To be noted, this study only analyzed the association between the RIL and the OS of EC patients, and we found there were some related studies that were not included in this meta-analysis [5,6]. Therefore, we performed this more comprehensive systematic review and meta-analysis.

Point 5: In the discussion section, paragraph 1 authors state that “Compared with the previous meta-analysis which only included 3 cohorts and found no significant association between RIL and the OS of EC patients, our systematic review was much more comprehensive”.

Please restate this statement after revision of your interpretations from the cited meta-analysis.   

Response 5: Thank you for your comments. As you required, we have now added new sentences in the discussion section, paragraph 1 as follows:

“Compared with the previous meta-analysis [1] which only included 3 cohorts and found a trend toward statistical significance that RIL was associated with inferior OS of EC patients, our systematic review was much more comprehensive, which improved and expanded their findings.”

Specific comments:

Point 6: A space is required between the end of the statement and the bracket, including the citation numbers.

Response 6: We have now revised it.

Point 7: Authors often used the word “would” in the text. For example, “The meta-analysis would be conducted with equal or over 3 individual results……”. Authors are advised to use appropriate tense or must take help from professional writers. 

Response 7: Thank you for your kind suggestion. We have now carefully checked our manuscript and revised the potential language problems and added new sentences to make our manuscript more readable.

In addition, we also rephrased the conclusion with softer terms than in the previous manuscript version to make our conclusion more rigorous. We hope our revision could meet your concerns. Thank you again for your valuable comments.

References

  1. Damen, P.J.J.; Kroese, T.E.; van Hillegersberg, R.; Schuit, E.; Peters, M.; Verhoeff, J.J.C.; Lin, S.H.; van Rossum, P.S.N. The Influence of Severe Radiation-Induced Lymphopenia on Overall Survival in Solid Tumors: A Systematic Review and Meta-Analysis. Int J Radiat Oncol Biol Phys 2021, 111, 936-948, doi:10.1016/j.ijrobp.2021.07.1695.
  2. Deng, W.; Xu, C.; Liu, A.; van Rossum, P.S.N.; Deng, W.; Liao, Z.; Koong, A.C.; Mohan, R.; Lin, S.H. The relationship of lymphocyte recovery and prognosis of esophageal cancer patients with severe radiation-induced lymphopenia after chemoradiation therapy. Radiother Oncol 2019, 133, 9-15, doi:10.1016/j.radonc.2018.12.002.
  3. Zhang, E.; Deng, M.; Egleston, B.; Wong, J.K.; Su, S.; Denlinger, C.; Meyer, J.E. Dose to Heart, Spine, Aorta, and Body Predict for Severe Lymphopenia and Poor Survival in Patients Undergoing Chemoradiation for Esophageal Cancer. International Journal of Radiation Oncology Biology Physics 2019, 105, E206-E207, doi:10.1016/j.ijrobp.2019.06.2041.
  4. Xu, H.; Lin, M.; Hu, Y.; Zhang, L.; Li, Q.; Zhu, J.; Wang, S.; Xi, M. Lymphopenia During Definitive Chemoradiotherapy in Esophageal Squamous Cell Carcinoma: Association with Dosimetric Parameters and Patient Outcomes. Oncologist 2021, 26, e425-e434, doi:10.1002/onco.13533.
  5. Liu, M.; Li, X.; Cheng, H.; Wang, Y.; Tian, Y. The Impact of Lymphopenia and Dosimetric Parameters on Overall Survival of Esophageal Cancer Patients Treated with Definitive Radiotherapy. Cancer Manag Res 2021, 13, 2917-2924, doi:10.2147/CMAR.S297010.
  6. Wang, X.; Zhao, Z.; Wang, P.; Geng, X.; Zhu, L.; Li, M. Low Lymphocyte Count Is Associated With Radiotherapy Parameters and Affects the Outcomes of Esophageal Squamous Cell Carcinoma Patients. Front Oncol 2020, 10, 997, doi:10.3389/fonc.2020.00997.

Round 2

Reviewer 3 Report

Minor comment 1: In lines #128 and #129, the authors mention that "Several medicines, such as nivolumab and pembrolizumab, that block the connection between PD-1 and PD-L1 are now being utilized to treat EC [38].

Please re-word the statement using specific terms such as immunotherapy or immune checkpoint blockers instead of medicine which may be also interpreted as a drug molecule.

Author Response

Point 1: Minor comment 1: In lines #128 and #129, the authors mention that "Several medicines, such as nivolumab and pembrolizumab, that block the connection between PD-1 and PD-L1 are now being utilized to treat EC [38].

Please re-word the statement using specific terms such as immunotherapy or immune checkpoint blockers instead of medicine which may be also interpreted as a drug molecule.

Response 1: Thank you for your kind suggestion. We have now changed those sentences to “Several immune checkpoint inhibitors that block the connection between PD-1 and PD-L1 are now being utilized to treat EC, such as nivolumab and pembrolizumab [38]”.

Thank you again for your valuable comments, which greatly improved our manuscript.